# Inequality in economic shock exposures across the global firm-level supply network

Abhijit Chakraborty [1,7], Tobias Reisch [1,2,7], Christian Diem [1,3], Pablo Astudillo-Estévez [1,4,5] & Stefan Thurner [1,2,6] ✉

For centuries, national economies have been engaging in international trade and production. The resulting international supply networks not only increase wealth for countries, but also allow for economic shocks to propagate across borders. Using global, firm-level supply network data, we estimate a country's exposure to direct and indirect economic losses caused by the failure of a company in another country. We show the network of international systemic risk-flows. We find that rich countries expose poor countries stronger to systemic risk than vice-versa. The risk is highly concentrated, however, higher risk levels are not compensated with a risk premium in GDP levels, nor higher GDP growth. Our findings put the often praised benefits for developing countries from globalized production in a new light, by relating them to risks involved in the production processes. Exposure risks present a new dimension of global inequality that most affects the poor in supply shock crises.

Interconnected supply chains forming complex networks span the globe as a consequence of centuries of globalization[1]. International production and trade played an essential role in increasing economic growth[2–4], reducing global income inequality, especially due to above average growth in China and South Asia[5], and has been argued to positively affect sustainable development[6]. Previous studies also argued that outsourcing or delocalization abroad of domestic production can have negative effects. Globalization allowed firms to execute strongly polluting tasks abroad[7–10] and to outsource labor intensive tasks to countries with weaker labor-rights[11,12] or unsafe or violent working conditions[13–15]. International trade creates demand for and facilitates the spread of problematic goods such as conflict minerals[16]. Importantly, international trade relations also act as direct and indirect transmission channels for economic shocks, such as supply or demand reductions[17–20].

Several recent works show that production networks act as transmission channels for economic shocks. A study on natural disasters in the United States[21] finds substantial evidence that shocks propagate from suppliers to customers, with customers suffering an average output drop of 3.1% four quarters after the disaster. Similarly, shock spreading on the firm-level production network subsequent to the Great East Japan Earthquake 2011 has been estimated, using an Agent Based Model, to have caused value added reductions equivalent to 2.4% of GDP in the year after the earthquake, more than 100 times more than the immediate direct losses[22]. A similar study[23] finds that customers of firms affected by the same earthquake suffer of an average drop in sales of 3.8% in the following year. The propagation of these shocks has been estimated, using a general equilibrium model, to have caused a significant 0.47 percentage point decline in Japan's real GDP growth the year following the earthquake. Further, the same event also provided evidence for the cross-country spread of economic shocks. Using firm-level data[24] shows that close affiliates of Japanese corporations in the USA experienced large drops in output in the months following the earthquake.

So far, risks of international economic shock propagation are typically studied on highly aggregated flows of goods between countries[17–20,25]. However, the intricate topology of the firm-level global supply network has a potentially crucial role for how economic shocks spread across countries. The importance of knowing the detailed network topology for understanding the spreading of shocks has been

[1]Complexity Science Hub Vienna, Vienna, Austria. [2]Section for Science of Complex Systems, CeMSIIS, Medical University of Vienna, Vienna, Austria. [3]Institute for Finance, Banking and Insurance, Vienna University of Economics and Business, Vienna, Austria. [4]School of Economics, Universidad San Francisco de Quito, Quito, Ecuador. [5]Institute for New Economic Thinking, University of Oxford, Oxford, UK. [6]Santa Fe Institute, Santa Fe, NM, USA. [7]These authors contributed equally: Abhijit Chakraborty, Tobias Reisch. ✉e-mail: stefan.thurner@meduniwien.ac.at

shown extensively in the finance literature. Shock propagation mechanisms for identifying systemic risks in economic systems were first developed for financial networks, consisting of banks and liabilities between them[26–31]. The risk of a local disruption, e.g. the default of a single bank, causing a system-wide large disruption in a financial network (financial crisis) is called the *systemic risk* it poses to the system. A full assessment of the exposures created by a financial agent and thus its systemic risk contribution to the system is not just given by its size (i.e. the sum of its direct exposures), but depends crucially on its position in the network. A reasonable quantification of systemic risk involves the detailed knowledge of financial networks; A particularly practical quantity (network centrality measure) is the DebtRank[28,29,32,33] that relates the failure of a bank to the caused systemic losses.

Only recently, these ideas were applied and generalized to the real economy and supply networks[34]. In networks formed by the supply-demand interactions between firms, the default of a single company may cause—through cascading dynamics—disruptions in large parts of the system. A corresponding Economic Systemic Risk Index was developed specifically for production networks[35]. In a similar direction, agent-based-modelling approaches for estimating the economic cost of the failure of a group of firms, known as regional adaptive input output models, have been developed in the context of (natural) disaster impact assessment[22,36–38].

Works of this kind demonstrate that network effects on the firm level are relevant and are too large to be ignored. The standard approach of quantifying exposures is the input-output (IO) analysis on the sector-level. There, all networks with a resolution that is more granular than the sector level are ignored. Illustrative examples of shock spreading on the single firm level are the global shortage in hard disk drives subsequent to the 2011 flood in Thailand[39], or the global shortage in computer chips of 2021/2022[40,41]. In the wake of the COVID-19 crisis, the distress of one firm—*Taiwan Semiconductor Manufacturing Company, Limited*—lead to production interruptions and layoffs in companies on the other side of the globe, e.g. in European and US car manufacturers[42,43]. While several studies have addressed the structure of global economic networks[16,44,45], studies investigating economic shock spreading dynamics on the global firm-level supply network are few. A notable exception is[24] where US affiliates of companies affected by the Great East Japanese Earthquake are studied.

The underlying principle of the above approaches is the understanding of shock propagation on the underlying economic networks in combination with the corresponding actual economic mechanics. Note that there are important differences between financial and production networks. In the former links (assets and liabilities) are stock

quantities, while in the latter links represent traded goods and services which are flow quantities. In this work we focus on production networks and their systemic risks only. In Fig. 1a we show the situation for a cascading failure in an international production network subsequent to the failure of an initial node, firm 1, marked by the red cross. Because firm 1 will no longer produce any goods, it cannot supply inputs to firms 2 and 3, so they have to reduce their production as well and cannot supply to their customers (ignoring potential substitution effects). Iterating this logic leads to a new stable configuration of reduced production levels shown in Fig. 1a. The filling of the nodes mark the reduction in economic activity (production). In the case of a supply shock, we call this mechanism the *downstream* propagation of shocks or the *downstream cascade*. The same logic applies to demand reductions that propagate *upstream*, or equivalently, cause an *upstream cascade* (not shown). See for example in[46] for an analysis of which conditions cause either the *upstream* or *downstream cascade* to occur.

The argument that all involved parties benefit from international trade dates back to David Ricardo's theory of *comparative advantage*[47]. However, because trade links can act as channels for shock transmission in production networks, it exposes firms and countries to risks they usually can neither assess, nor control. Therefore, quantitative measures are needed that allow us to compare the benefits with the inherent production risks of a globalized economy. Here we develop a novel measure to quantify a country's exposure to economic shocks from international production and trade, based on a microscopic shock propagation mechanism. It is an estimate for the expected economic loss a country is exposed to if an arbitrary firm fails in another country. As such it is also a novel measure for a country's resilience to supply network shocks originating in other countries. For this, we define the direct exposure to be the production losses caused by a (temporary) production failure of a direct supplier (or customer) firm. Indirect —or, equivalently, higher order— exposures result from the propagation of the direct shocks along indirect supply relations (supplier of a supplier and so forth). In the following, the term 'exposure' refers to the sum of direct and indirect exposures. We can then quantify differences in the countries' risk exposures, and discover how these exposures are distributed across the globe. The estimated exposures can then be put into perspective with measures of gains from globalized production and trade. We focus on an accurate description of international economic shock propagation and compare the exposure to direct and indirect production losses with GDP growth, as a straight forward proxy for benefits from globalized production and trade. With this study we provide a first step towards an understanding of the global flow of systemic risk and how it is

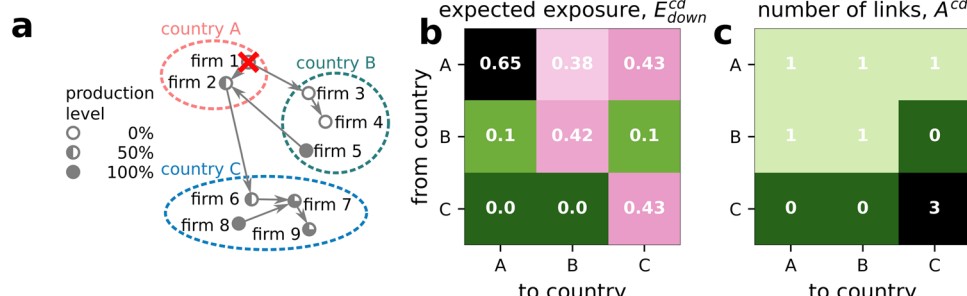

**Fig. 1 | Shock propagation on a firm-level global supply network. a** A cascade of production reduction in a toy economy of three nations and nine firms, subsequent to the default of firm 1, marked by a red cross. The filling of the nodes (pie chart) indicates their remaining production level after the shock propagated. The production of a firm is reduced proportional to the relative amount of inputs not available. **b** Exposures to production losses between the three countries from single firm failures. The expected downstream exposure country *c* poses to country

*d*, $E_{down}^{cd}$, is measured as the average production loss in country *d* if a random firm in country *c* defaults. Green (magenta) indicates low (high) exposure to expected production losses. **c** Number of links between two countries, $A^{cd}$. Green (magenta) indicates low (high) number of direct links. The country-adjacency matrix $A^{cd}$ is a poor predictor of $E_{down}^{cd}$, highlighting the relevance of the of intra-country network topology.

distributed across the globe. Together with new records of global supply network data this will open the door to reliably assess who carries the burden of the indirect economic costs caused by future crises, such as natural disasters or pandemics.

We use a global firm data set, see Materials and Methods, as a starting point to reconstruct an international firm-firm supply network. We use it to compute a number of different exposures, based on the concept of DebtRank, see SI Text 1. While DebtRank $R_i$ in its basic form quantifies the systemic relevance of a node by aggregating the losses it could cause in the entire network, here we generalize it such that the systemic relevance of firms to specific countries can be quantified. We thus adapt the cascading mechanism underlying $R_i$ on supply networks in[34] to quantify the effect of the failure of each single firm on every national economy.

Our approach is agnostic to the economic shocks the firms in the source countries suffer. We are assuming a complete failure and removal of a firm from the production network, to subsequently study the emerging disruption chains *ceteris paribus*. We simulate the spread of short term shocks, until all firms that could be possibly affected have been affected, and there are no shocks propagating anymore. By allowing no recovery and shocks to spread across the network we associate a firms' systemic risk with the worst case estimate of economic damage it can do. This is of course a simplification of the real situation, and previous research has shown that delays, substitution dynamics, and other non-linearities can change the size and time scales of the emerging cascades[46,48]. However, employing more sophisticated models would require more assumptions, because, due to a lack of data, we cannot reasonable estimate the time scales and inventory levels of the firms in our network. Instead, we're using a slighlty simpler model and focus on the role of the network structure. In the main text, we present the results of shocks propagating based on the assumption that the impact of a supplier failure on a customer is proportional to its input share. In SI Text 2 we test the robustness of our results by introducing a simple replacement dynamics mechanism that softens the impact of a supplier failure.

We define the following quantities: The *Country-Firm Exposure*, $E_i^d$, of country $d$ to firm $i$, quantifies the fraction of the national production of country, $d$, that is lost in case of firm $i$'s failure. This includes direct and indirect shocks transmitted through the network; for details, see Materials and Methods. We define the *Country-Country Exposure*, $E^{cd}$, as the average exposure of a country $d$ to production losses originating from random firm failures in country $c$. Assuming that we pick a random firm in country $c$, $E^{cd}$ amounts to the average over all $E_i^d$, $E^{cd} = \sum_{i \in \mathcal{C}^c} E_i^d / |\mathcal{C}^c|$, where $i$ is located in country $c$ and $|\mathcal{C}^c|$ is the number of firms in country $c$. Note that the average exposure, $E^{cd}$, measures the robustness of country $d$ to shocks from country $c$, low (high) values imply high (low) robustness. $E^{cd}$ measures a country's relative production reduction. For absolute exposure values, we define a country's *Country-Country Exposed Value*, $V^{cd}$, by multiplying the relative average exposure with the economic size $k^d$ of the affected country, $V^{cd} = k^d E^{cd}$. We approximate a firm's size by its number of links, $k_i$, and the economic size of a country by the sum of its firm-sizes, the total degree, $k^c = \sum_{i \in \mathcal{C}^c} k_i$. To distinguish between up- and downstream effects, we define the above measures for both up- ($E_{up}^{cd}$) and downstream ($E_{down}^{cd}$) cascades separately. We define the total (imported) exposure as $E^d = \sum_{c, c \neq d} E^{cd}$. Note that $E^{cd}$ is conditional that a random firm in $c$ fails. Assuming that the default probability for firm $i$ is $p_i$, we can also calculate the *expected loss* $\tilde{E}^{cd}$, it is calculated by forming the expectation values over all production losses, $\tilde{E}^{cd} = \sum_{i \in \mathcal{C}^c} p_i E_i^d / |\mathcal{C}^c|$[33]. Lacking information on firm default probabilities, we discuss the *average* exposure $E^{cd}$ in the main text and discuss $\tilde{E}^{cd}$ based on simulated PDs in SI Text 3.

We exemplify the quantities by calculating them for the toy example in Fig. 1a. The default of firm 1 causes production in country B to drop by $E_1^B = 75\%$. The default of firm 2, as shown in SI Fig. S2, SI

Text 4, has no effect on country B, $E_2^B = 0\%$. If we randomly pick a firm in country A, now the expected country exposure is the average of the two firms $E^{AB} = (0 + 0.75)/2 = 0.375$. Figure 1b shows the $E_{down}^{cd}$ matrix for the toy network shown in panel a; dark magenta means high, green means low exposures, respectively. Country A creates the most risk, affecting B and C; C creates the lowest, affecting only itself. For every row the highest value is found in the diagonal, highlighting that firms within countries expose each other stronger among themselves than between countries. The comparison of $E_{down}^{cd}$ in Fig. 1b with the number of links between countries $A^{cd} = \sum_{i \in \mathcal{C}^c j \in \mathcal{C}^d} A_{ij}$ in Fig. 1c indicates that the systemic risk spreading dynamics contains a lot of effects that are not visible when only considering the links between countries. Both for directly and indirectly connected countries, much of the heterogeneity in shock spreading is lost when ignoring the firm-level network topology. The information loss by aggregation is best seen when comparing different realizations of the firm level supply network, $A_{ij}$, that have the same country-level aggregation, $A^{cd}$. For example, if we rewire the outgoing link of firm 5, such that it now supplies firm 1 instead of firm 2, we preserve the number of links between countries $A^{cd}$, but the exposure of countries A and C to B would be substantially larger than compared to the situation shown in Fig. 1, see SI Fig. S6 in SI Text 4. For a detailed study of the effects of aggregation on the mis-estimation on shock spreading dynamics, we refer to[49].

## Results

First we analyze the network structure of the average exposure between countries, $E_{down}^{cd}$. We find that *country country exposures* cluster in geographic regions. We see this by sorting the countries according to regions and continents in Fig. 2. Figure 2a shows $E_{down}^{cd}$; panel b shows the number of firm-firm links, $A^{cd}$, between countries $c$ and $d$. Quantities are in logarithmic scale. The strongest connectivity and exposure is found within countries, as seen in the high values along the diagonal in the $A^{cd}$ and $E_{down}^{cd}$ matrices. The effect is much stronger for $E_{down}^{cd}$ than for $A^{cd}$.

Economic exposure can be expected to be strong between countries in the same geographic or economic region, such as the European Union or the United States-Mexico-Canada Agreement. Figure 2a $E_{down}^{cd}$ clearly shows a corresponding block-diagonal structure, highlighting closely connected regions on all continents, even differentiating between sub-regions, such as northern and sub-Sahara Africa. Further prominent blocks are seen in the middle east and south Asia, eastern Europe, in contrast to northern, southern and western Europe which expose almost all countries, and South America. The magenta colored horizontal lines in Fig. 2a indicate that some countries export downstream exposures to almost all other countries. They are particularly notable for a block of western European, several Asian and North American countries. Rich economies appear to expose countries beyond their own region. For each block we perform a one-sided Mann-Whitney U test to confirm that exposures to countries within the same continent are indeed larger than to countries outside. The tests are significant for all continents at a $p < 0.01$ level. For a more detailed visualization and discussion of $E_{down}^{cd}$, containing single country indices, see SI Fig. S7 and SI Text 5. For $A^{cd}$, presented in Fig. 2b, these structures are only barely visible, except faintly for parts of Europe (EU), Asia (AS) or South America (SA). In SI Text 6, we investigate this aspect further by comparing the exposed value $V^{cd}$ to the average number of links from firms in country $c$ to country $d$, $\bar{k}^{cd}$. Although $V^{cd}$ and $\bar{k}^{cd}$ are highly correlated (Pearson's $r = 0.93$, $p < 10^{-15}$), we find variations of up to two orders of magnitude in $V^{cd}$ for a given $\bar{k}^{cd}$, see SI Fig. S8.

The obvious asymmetry in $E_{down}^{cd}$ in Fig. 2a suggests large differences in how systemic risk is distributed around the globe and that exposure to economic losses is usually not reciprocal. In Fig. 3a we change the aggregation scale to three income groups, containing the

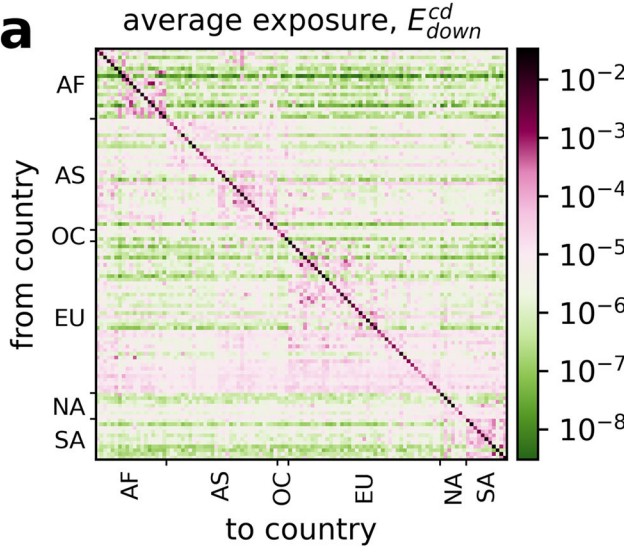

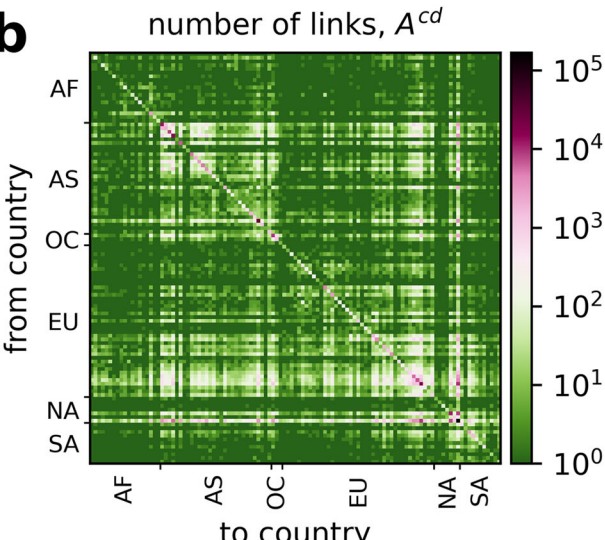

**Fig. 2 | Exposures between countries in the international supply network, aggregated to the country level. a** Expected fraction of the economy affected in country $d$, following a (random) firm default in country $c$, $E_{down}^{cd}$ (in logarithmic scale). Lines and columns are sorted by continent and region, Africa (AF), Asia (AS), Oceania and Australia (OC), Europe (EU), North America (NA), and South America (SA). Exposures are highest along the diagonal. One can clearly see a regional block structure indicating large exposures within regions and economic blocks. This is

especially visible for AS, SA, AF. We find prominent horizontal lines (magenta), indicating that some countries create exposure well beyond their geographic region, exposing almost all other countries. A larger representation with all values and individual country numbering can be found in SI Fig. S2. **b** Country-level adjacency matrix, $A^{cd}$, showing the number of firm-firm links from country $c$ to country $d$, in the same order as in **a**. To avoid problems with the logarithmic scale, we add 1 to every entry before taking the log.

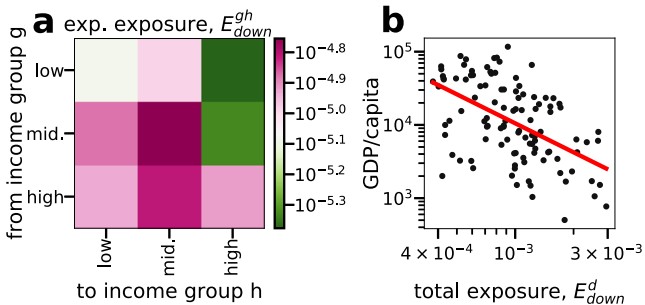

**Fig. 3 | Exposures between high-, middle, and low income countries. a** Exposure (log-scale) between firms separated into low-, middle- and high-income groups based on their country's GDP per capita. Firms in middle-income countries experience the largest amount of exposure and high-income countries are mostly exposed to other high-income countries. Firms in low income countries are much more exposed than they expose others. SI Table S1 lists the income group for each country. **b** GDP per capita plotted against total exposure, $E_{down}^{d}$. A significant negative correlation of $r = -0.52$, $p < 10^{-8}$ highlights that higher exposure is connected to lower income per capita. The red line represents the log-log ordinary least squares regression fit to the data of form $y \sim x^{-1.31}$.

same number of countries, based on their GDP per capita –low, middle and high. We plot the respective $E^{gh}$, where $g$ and $h$ denote the respective country income groups. Three facts become obvious. First, firms in high income countries cause, on average, much more distress to middle and lower income countries than the other way round. Second, firms in high and middle income countries are most exposed to firms in countries from their own respective groups. Firms in low-income countries receive relatively little risk from their own income group but more from middle and high income countries, indicating that these countries' economies are more exposed to the wealthier trading partners. Third, the dark color of the column for middle income countries in Fig. 3a indicates that they are exposed to most of the risk. In SI Text 7 we fit gravity models for $A^{cd}$ and $E_{down}^{cd}$, respectively. We find that the number of links between countries, $A^{cd}$, is well

explained by a classic gravity model, but the *Country-Country Exposure*, $E_{down}^{cd}$, is not well explained. We find that $E^{cd}$ depends only weakly on the exposure created by $c$ and inversely on the exposure received by $d$, emphasizing that countries receiving a lot of exposure do so not through a few large exposures, but many small.

In Fig 3b we plot the total exposure, $E_{down}^{d}$, versus GDP per capita showing an anti-correlation (Pearson $r = -0.52$, $p < 10^{-8}$). This indicates that countries with a low GDP per capita are more exposed than countries with a higher. To control for confounding factors, we perform a multivariate linear regression, where GDP per capita is the dependent variable and $E_{down}^{d}$, GDP, imports, and exports are the independent variables, all variables are log-transformed. The results of the regression analysis are shown SI Tab. S3 and SI Text 8 and indicate that in the presence of these control variables $E_{down}^{d}$ has a statistically significant influence on the dependent variable. The model explains 50% of the variance and is highly significant (adjusted $R^2 = 0.50$, $p < 10^{-15}$).

Generally, high risks are thought to be compensated with higher returns. We check for the existence of these by comparing $E^d$ with the average annual growth of GDP per capita in the respective country over the past twenty years. Similarly, a high growth rate has been causally connected to successful export strategies[50,51]. SI Figure S10, in SI Text 9, shows that there is little to no correlation between the variables (insignificant Pearson correlation of $r = 0.15$, $p = 0.12$). Countries seem not to be compensated for taking systemic risk. In SI Text 9 we check for the robustness of this result by comparing with the average annual GDP growth over 5 and 10 years, as well as the average net inflow of foreign direct investments over 20 years.

To find out how equally exposures are distributed across the globe, in Fig. 4 we plot the respective Lorenz curve (shown in red), i.e. the cumulative share of population (sorted by their countries' $E_{down}^{d}$ per capita) versus their cumulative share of global average exposure, $E_{down}^{d}$. In total, the 80% of population that are least exposed to risk are exposed to only around 10% of all risk, or vice-versa, the 20% most exposed population carries around 90% of the risk. For comparison, in Fig. 4 we also show the Lorenz curve for the GDP (blue). Clearly, GDP is distributed more equally than the $E_{down}^{d}$, which is also shown by the

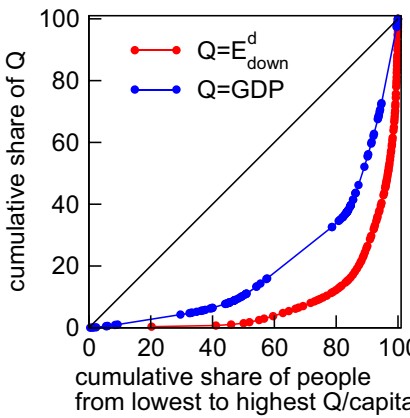

**Fig. 4 | Lorenz curves for total exposure and GDP of all countries.** The red (blue) line shows the proportion of risk (GDP) that is associated to the lowest exposed (poorest) x% of people globally. For perfectly equally distributed exposures (wealth) the curve would coincide with the diagonal. Exposure to economic shock is obviously distributed more unequally than GDP. We find Gini coefficients of 0.83 and 0.59 for $E^d_{down}$ and GDP, respectively. Systemic risk exposures represent a new aspect of inequality.

Gini coefficient. We find Gini coefficients for the GDP and total exposure $E^d_{down}$ to be 0.59 and 0.83, respectively.

So far we only considered *downstream shock propagation* by simulating supply shortages. The analysis for the *upstream* cascade is discussed in SI Text 10. We find similar structures with respect to the expected upstream exposure $E^{cd}_{up}$ matrix (analogously defined as $E^{cd}_{down}$) in SI Fig. S6. The distribution of upstream exposure by income group in SI Fig. S7 reveals a different picture than for downstream risk. Most risk is between low- and middle-income countries and high-income countries neither create nor are exposed to large amounts of risk. The respective Gini coefficient is 0.81 and therefore higher than for GDP, again to the disadvantage of poor countries (Pearson's correlation between GDP per capita and $E^{cd}_{up}$ is $r = -0.20$, $p < 0.04$, see SI Fig. S8.

Here we only presented the results for the *average exposure*, $E^{cd}_{down}$. In SI Text 3 we discuss the *expected loss*, $\tilde{E}^{cd}$, and show the robustness of our results to heterogeneity in the failure probabilities of the firms.

## Discussion

We present a method to quantify the exposure of countries to production losses caused by firm defaults in other countries based on global firm-level supply network data. We introduce the *average Country-Country Exposure*, $E^{cd}$, as the average relative loss of production in country $d$ after a random firm-failure in country $c$. The method enables us to demonstrate that exposures to other countries are highly structured on a regional level, and that high income countries expose a large fraction of the globe to economic losses. Low income countries are disproportionately strongly affected by high exposure values. Somewhat contrary to intuition, it seems that higher exposure is not positively correlated with higher gains in GDP growth rates in recent decades. Global economic exposure of the type discussed is distributed more unequal than income per capita.

The presented metric confirms the intuition that average exposures of countries are highest to firms within their own economies, and on the international level exposures are strongly influenced by geography. When countries are sorted by continent, the country exposure matrix, $E^{cd}_{down}$, shows a prominent block-diagonal structure. This is not unexpected, since it is known that exposure increases with connectivity, see SI Text 6, and because trade intensity decreases with distance[52]. Several wealthy countries create significant exposures

beyond their region and thus export systemic risk to other countries. To make sure this is not due to a size effect, $E^{cd}$ is normalized by the number of firms in country $c$. However, there could be a remaining size effect, because a country's total degree does not scale linearly with its number of firms. In SI Text 11 we show that our results remain qualitatively the same if we normalize with the country's total degree instead. Note that the values for $E^{cd}$ are small, but represent the average exposure per firm we expect in country $c$ after a random firm failing in d. On the one hand, these numbers can get large if there would be, for example, a systemic event in country $c$, affecting many firms. On the other hand, large values in $E^{cd}$ are often driven by few firms causing large cascades in $d$.

To understand if the asymmetry of few countries exposing many countries and many countries exposing only a few is related to their average income level, we separate countries into low-, middle-, and high income groups. We find that high income countries are only exposed to other high income countries, while low- and middle income countries are exposed to shocks from firms in countries of all income levels. The asymmetry of exposure does a priori not provide any indication whether rich or poor countries have higher total exposure. On the one hand, poor countries could be more dependent on inputs from rich countries, on the other hand rich countries could depend more on highly integrated global value chains and, hence, be more exposed to systemic risk. A regression analysis solves the puzzle and we find that GDP per capita correlates negatively and, hence, poor countries have a higher total exposure. We find no evidence for a "risk premium" in the sense that higher exposures do not co-occur with significantly higher economic growth rates. The average exposure is not significantly correlated with the growth of GDP per capita during the past 20 years before the COVID-19 crises.

We investigate the inequality of exposures across the globe and find that exposure is highly concentrated with a Gini coefficient of 0.83 (compared to 0.59 for GDP). This additional dimension of global inequality adds to a number of other forms of inequality such as inequality with respect to health, (formal) education, and wealth[53]. Inequality impedes economic growth[54,55] and has been associated with being one of the driving factors for the collapse of societies[56]. In their Agenda 2030 the United Nations set to "reduce inequality within and among countries" (SDG 10) as one of the 17 sustainable development goals (SDGs)[57]. Future research should investigate if the network structure of the global supply network locks in already existing inequalities. For social networks it has been shown that their structure can procreate inequality[58,59]. Similarly, our results suggest that structural inequality in production and trade networks between countries wrt. systemic risk exposure is significant and should be taken into consideration in future efforts to fight inequality in its various forms. Since exposure inequality arises from the structure of the supply network on the firm-level, it is important to understand the processes that let firms from different income countries enter into production and trade relations and how this could happen with creating less risk exposure to poorer countries. Strategies of incentivizing firms to become systemic risk sensitive could be a starting point[60,61].

The presented work shows how to calculate international direct and indirect economic exposures between countries that is methodically based on previous work on financial and supply networks[28,34]. The present work is only a first step in the direction of quantifying systemic risk flows around the globe and has several limitations that need improvement. First, our approach disregards completely the nature of the products. An accurate description of shock propagation must include the type of the goods and their firm-level production functions. Only then can a supply network be considered a realistic production network. At the moment, the shock spreading dynamics are practically limited to the special case of linear production functions that are typically underestimating the actual shock spreading risks[35,49]. Second, the global supply network used in this study comprises

$N = 230, 970$ firms, two orders of magnitude less than the likely number of firms in the world[62], thereby introducing a potential sample bias. In SI Text 12 we investigate this potential bias by sampling nodes from the global supply network and re-running our shock spreading algorithm. We find a high correlation between the results on the full and the sampled networks, and can qualitatively reproduce our results. Third, the current dataset lacks information on the firm's revenues and the traded volumes. In SI Fig. S23, in SI Text 13, we show that GDP, imports, and exports all have a high correlation with a country's total degree, outdegree, and indegree, respectively. Moreover, we test the robustness of our results when omitting low-volume links and ignoring the relative importance (volume) of links, using a firm-level supply network that is reconstructed from high-quality value added tax data from Ecuador. In SI Text 12 we show that our results are robust with respect to the removal of low-value transactions and ignoring link-weights. We also show that in Ecuador a firm's degree is highly representative of its sales, especially when discarding low-value links. Nevertheless, more detailed supply network information would of course improve the quality of the results since relative effects in the supply network propagation would be correctly captured. Fourth, the dataset is compiled by firms focused on the US, entailing a potential reporting bias. To ensure the robustness of our results, we perform the same study for states in the USA. Although the heterogeneity in the US is much lower, we find similar trends, such as high inequality to the benefit of rich states, for details see SI Text 14. Fifth, the shown results were derived for the spreading of supply and demand shocks separately. We find qualitatively similar results; there is high inequality to the disadvantage of low income and advantage of high income countries. However, it would be desirable to design a measure that is able to capture up- and downstream risk simultaneously, similar to the ESRI quantity recently presented in[35]. Sixth, our model runs the shock spreading mechanism on the empirical network until it converges, not taking into account inventories, substitution dynamics, or firm recovery capabilities. In SI Text 2 we attempt a partial validation of our model by comparing the impact a supplier failure has on its customers with values from the literature. Further, to test the robustness of our results, we introduce a mechanism for input substitution and find that the patterns described in the results persist. Seventh, the results presented in the main text do not account for default probabilities of firms, $p_i$. This is certainly unrealistic, and we show that our results are robust to heterogeneous $p_i$ by including random PDs in SI Text 3. In further work, empirical, heterogeneous default probabilities of companies should be taken into account, as well as information on correlated shocks, e.g., due to natural disasters or climate change. However, this is presently practically impossible due to unavailability of corresponding data. Eighth and finally, we are not able to perform a full validation of the general results of our shock spreading cascade, in particular $D_{ij}$ and $E^{cd}$. This is mostly due to a lack of data and beyond the scope of this work, but would in principle be possible. Such efforts would follow[21,23,24] and investigate changes in revenue of firms (or countries) after exogenous shocks affecting single or multiple firms.

The presented work allows us to derive three immediate policy implications: First, we conclude that the (global) distribution of exposure to economic shocks must be traced on granular representations of the underlying networks. For this a global effort on collecting and monitoring the according data is necessary. This would allow to anticipate and prepare for globally spreading supply shocks. Second, since inequality is deeply embedded in the economic structure of the global supply network, future efforts to reduce inequality, such as the goals formulated in SDG 10 in the United Nations' Agenda 2030, must include the systematic restructuring of the global production network to spread exposure risk more fairly. Third, since the developed index quantifies the spread of exposure to economic shocks between countries. The employed firm-level resolution allows us to straight forwardly adapt systemic risk management methods for national economies. A possible incentive scheme could be an appropriately adapted systemic risk tax[60,61] for international supply networks.

## Methods
### Data
We use supplier-customer data from the web site of Standard & Poor's (S&P) Capital IQ platform for the year 2017. The data is comprised of firm identification (ID), name, location, primary industry, and sector as node information. Using the Global Industry Classification Standard (GICS), Morgan Stanley Capital International and S&P grouped the firms into 11 sectors and 158 primary industries. Firms are distributed over 206 countries. The data contains information on 1,403,807 business relationships between firms, such as supplier, creditor, franchisor, licensor, landlord, lessor, auditor, transfer agent, investor relations firm, and vendor. Most relations are of type supplier or creditor (69% supply links, 31% creditor links. The supplier type implies that a firm provides goods or services to other firms, the creditor type indicates that a firm lends money to other firms. For a detailed description of the dataset we refer to[45]. Because we are interested in the flow of goods and services between countries, in the following we restrict our analysis to the 968,627 supplier relations.

We preprocess the data in the following way. We remove all firms that do not have locations or sector classification information. To avoid misleading results from countries with too few firms, we only consider firms from those countries where the total number of firms exceeds 30. Further, we remove firms from Barbados, Bermuda, British virgin Island, Channel Island, Gibraltar, Monaco, and Cayman Island as these are known for having considerable numbers of offshore firms. We construct an unweighted network with an adjacency matrix, $A_{ij} = 1$, if there exists a link between firm $i$ and $j$ and $A_{ij} = 0$, otherwise. After removing isolated nodes, parallel links, and self-loops, the network contains $N = 230, 970$ firms and $L = 660, 701$ binary, directed links. For a collection of descriptive statistics of the network, see SI Text 15.

To explain the variation in the amount of distress propagation for different countries, we investigate the relation between distress and certain economic indicators characterizing an economy. We collect data on per capita gross domestic product, per capita exports and per capita imports for the year 2018 in current U.S. dollars from the website of the World Bank https://data.worldbank.org.

### Firm-country exposure
DebtRank, $R_i$, is a network centrality measure, initially developed as a financial systemic risk index for investment networks[28], and recently adopted for supply networks[34]. In a real economic network $R_i$ corresponds to the overall reduction in economic activity subsequent to the default of an initial firm $i$ and a possible cascade of defaults. Here, we adopt the specific underlying cascading mechanism to design a novel country-level indicator. In SI Text 1 we give a brief review of the DebtRank as used in[34] and here we describe our specific adaptation.

An intermediate result in the calculation of $R_i$ is the matrix $D$, with elements $D_{ij}$, denoting the distress firm $j$ receives if firm $i$ defaults (the loss in economic production firm $j$ experiences if firm $i$ defaults), see SI Text 1. The column $j$ contains the shocks firm $j$ is exposed to. Row $i$ contains the shocks firm $i$ causes to all other firms $j$ when it defaults. We use $D$ to define the effect of a node's default on a set of nodes $\mathcal{C}$, in our study typically corresponding to the firms in a country $c$, $\mathcal{C}^c$. In the following, lower indices denote nodes (firms) and raised indices denote groups of nodes (countries).

Because the global supply chain network lacks information on link-weights and node-sizes, we assigned each edge equal weight and associate the node-size $q_i$ with it's degree $q_i = k_i$, where $k_i$ represents the degree of the $i$-th node. The degree $k_i$ was chosen as a size proxy to be consistent and self contained in the network setting of the used dataset. However, one could also use any other size proxy such as

value added, turnover or employees. We define the weight of a country $c$ as the sum of its node weights

$$q^c = \sum_{i \in \mathcal{C}^c} q_i.$$

The *Firm-Country Exposure* of node i for country c is

$$E_i^c = \frac{\sum_{j \in \mathcal{C}^c} D_{ij} q_j}{q^c}.$$

Please note that the weight of the initial node is included in the denominator if it is also contained in the set $\mathcal{C}^c$. This allows the interpretation of this value as the fraction of the economy affected in country $c$ after the default of firm $c$.

### Reporting summary

Further information on research design is available in the Nature Portfolio Reporting Summary linked to this article.

## Data availability

The data used in this paper are proprietary and cannot be shared. However, they are commercially available through Standard & Poor's Capital IQ platform (https://www.capitaliq.com). Source data are provided with this paper.

## Code availability

The code for running the cascade and aggregation of its results is available on https://github.com/treisch/inequality_in_shock_spreading. Figure source data are provided with this paper.

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

## Acknowledgements

We thank Y. Ikeda for providing the S&P Capital IQ data. The authors thank the Centro de Estudios Fiscales of Ecuador's Servicio de Rentas Internas (SRI), which provided the Ecuadorian data for research purposes. The project was supported by Austrian Science Fund FWF under I 3073-N32, Austrian Science Promotion Agency FFG under 857136, and Hochschuljubiläumsstiftung of the Austrian National Bank OeNB under P17795 2018-2021.

## Author contributions

T.R., C.D., A.C., and S.T. conceived and designed the study. A.C. analyzed the data. P.A.-E. prepared the Ecuador data. All discussed results and contributed to the manuscript. A.C., T.R., and S.T. wrote the paper.

## Competing interests

The authors declare no competing interests.
