## [Peer Review File · Nature Communications]

Inequality in economic shock exposures across the global firm-level supply networkREVIEWER COMMENTS

Reviewer #1 (Remarks to the Author):

I read with great interest the paper "Inequality in economic shock exposures across the global firm-level supply network" by Chakraborty et al.

I believe this paper is a novel and interesting example of how the recent methodological advancements in complex network theory can be fruitfully applied to the specific case of study of economics.

The paper analyses the structure of the chain of production at various levels culminating in the structure of production at the continental size. Similar to previous work done for financial systems they are able to measure the risk exposure across the different levels.

In my view, the main result of the paper is that such exposure cannot be reasonably computed without taking into account the network effects in the system. Similarly to the financial case, non-considering such an effect will lead to a serious underestimation of the risk.

This paper fills a gap in the study of economic systems where network effects in the case of chains of production have been not fully considered. The methodology used is sound and reproducible, I have one question though when the authors consider the regional block structure in Fig.2.

One typical model for such things is the gravity model so that countries and/or firms have a tendency to trade with counterparts nearby so that one can consider the distance between the two places as an approximate measure of the unlikeliness of establishing a business relationship. It seems from the data (I might be wrong) that if it were so there should be different "gravity models" for any continent, making therefore such hypothesis not very plausible,

Could the author comment on this specific point, that is of great importance for the community?

For the rest I found the paper novel, well written and valuable as a contribution in the field and I have no hesitation in suggesting publication for it

Reviewer #2 (Remarks to the Author):

Review report

Title: Inequality in economic shock exposures across the global firm-level supply network

The paper investigates an important and extremely timely questions in the current fastly evolving geopolitical context. The paper is very well written and clear. The methodology is solid and transparent. I have some points of concern, as described in general comments. I believe addressing these points is feasible. However, it requires some substantial work.

General comments:

1. given that this is not an empirical analysis of losses actually occurred but results are based on a model of possible realizations of losses, then the authors should convince the reader why the relation highlighted in the paper between expected loss and network position and/or GDP per capita can be considered correct or relevant, or why/how is it useful from a perspective of scientific understanding and validation. I do think that the exercise carried out by the authors is useful, but I also think it is necessary to explain clearly why this is the case.

2. The analysis uses empirical data of supply networks. However, the magnitude of the effects estimated depends highly (as far as I can understand) on non-validated parameters (e.g. how much important a supplier loss is for a given company). Indeed, in absence of frictions, delays, inertia or other, a company could recover 100% from the shock in very short time. The cascades that the authors find only emerge, I suspect, in presence of delays or non-linearities. This was the result from Battiston et al. 2007 and I would be surprised there can be a different result here. Hence, the effects that the paper estimates entirely depends in magnitude on the recovery capabilities, which could be

measured empirically, in principle.

I suggest the authors to discuss how to carry out a validation of the model. Even if a full validation is not necessary and out of scope here, a partial validation could be attempted using data on the extreme/catastrophic events mentioned in the introduction. Another piste I suggest is to try and use existing empirical results to validate the microscopic mechanisms at work in this model. What matters is to convince the readers that the model could be validated and that it gives reasonable magnitude of estimates in a sample for which data is available.

If the success of the validation exercise is limited, a sensitivity analysis should be conducted to show how much the results are dependent on parameters related to the ability of firms to recover and also wrt to the PD of the firms. Currently, the PD is assumed the same for all firms, which is not very realistic. In reality this is going to be sector specific and time varying. What of these are details and what is important for our understanding?

3. The results for Ecd in Figure 2 seem smaller than 10^{-2} . These are small numbers. Why does it matter in economic terms?

How do these figure depend on assumptions? See point 2.

4. Related to point 2: I would suppose that the results are quite sensitive to how the parameters of the model are calibrated, e.g. the level of vulnerability of firms to the shock of disruption of one supplier. If this is the case, then what is the value of the results? Is the magnitude what matters here or something else? What is robust to those calibration details that we can learn from the paper?

Specific comments

Comment 1. Passage "Previous studies also argued that international production can have negative effects.". Wording: the term "international production" seems not correct or unclear. Authors may be mean the "outsourcing or delocalization abroad of domestic production".

Comment 2. Passage "In a similar study, the effect of the Great East Japan Earthquake has been estimated to have caused a significant 0.47 percentage point decline in Japan's real GDP growth [23]. " This sentence consider the same event as the previous sentence but difference between numbers is unclear. I suppose, 2.4% drop is a short-term shock, which this is a persistent reduction of growth rate. Please clarify. Please also address the fact that the expression "Great East Japan Earthquak" is repeated three times in one paragraph.

Comment 3. Introduction: The stream of works on the ambiguous effects of production networks could be put into the broader context of the global architecture of economic networks. In this regard, the first large scale empirical study of the global economic network is Vitali et al. 2011 PLOS-ONE, although it covers a different type of networks, namely ownership, which is of course different but also related to production networks. One similarity between the two papers is that they look at firm level networks as opposed to sector-level.

Comment 4 on passage "The potential of a local disruption, e.g. the default of a single bank, causing a system-wide large disruption in a financial network (financial crisis) is called its systemic risk.". I suppose the authors mean to say that this is its systemic impact not risk. A systemic event is usually an event with large losses at the system level. The authors should make consistent definitions of the expressions involving "risk". If risk is just a probability, then use probability instead. If "risk" is an expected loss please explain. In financial markets "risk" means volatility. So "risk" means different things to different audiences and it must be defined.

Comment 5 on passage: "Illustrative examples of shock spreading on the single firm level are the global shortage in hard drives subsequent to the 2011 flood in Thailand [39], or the ongoing shortage in computer chips [40, 41]." Please replace ongoing with a reference to 2022, for future readability.

Comment 6 on passage: "In the case of a supply shock, we call this mechanism the downstream

propagation of shocks or the downstream cascade. The same logic applies to demand reductions that propagate upstream, or equivalently, cause an upstream cascade (not shown).” This is indeed a very relevant concept, which was previously studied in Battiston et al 2007 in the context of a model of production network (coupled with credit). The authors showed conditions (analytically under homogeneity assumptions) for the occurrence of either one type of cascades or both.

Comment 7 on passage: “For example, if we rewire the outgoing link of firm 5, such that it now supplies firm 2 instead of firm 1, we preserve the number of links between countries A and C, but the exposure of countries A and C to B would be substantially larger than compared to the situation shown in Fig. 1.” Not very clear. The figure refers to the case with no rewiring. How do we see that after rewiring the difference would be big. The concept is important and probably worth showing explicitly.

Comment 8 on Fig 2. The term “Expected exposure from country ... to country” : the concept is clear but there is an English problem I suppose the common expression is exposure of a firm/country to another firm/country. I guess you can say impact from A to B but you cannot say exposure from A to B.

Comment 10 on Passage: “80% of down population that are least exposed to risk are exposed to only around 10% of all risk, or vice-versa, 20% of the most exposed population carries around 90% of the risk.” The passage could be revised to make easier reading the figure, e.g. referring to color of curve.

Discussion

Comment 11 on passage: “A regression analysis solves the puzzle and we find that GDP per capita correlates negatively ($r = 0.52, p < 10^{-8}$) and, hence, poor countries have a higher total exposure.” . I am confused now, is the passage referring to the same regression presented earlier in Results, or is it a new exercise?

Comment 12 on passage: “With the currently available data we find no evidence for a “risk premium” in the sense that higher exposures would co-occur with significantly higher economic growth rates.” The concept of risk premium appearing here is not explained. The term is used in finance and economics to indicate a differential in yield or return that market participants demand from holding a security in compensation for its risk. The analogy here is not clear and needs to be explained. I am not convinced it is a necessary or useful metaphor, since there is no holder of a contract here that could demand compensation for risk.

Comment 13. The last two pages of discussion are a bit confusing to me. They overlap with the section results and repeat some of the results. I suggest to streamline and remove redundancy. The section could be shortened substantially, unless I missed that there is important new information and then maybe there is a presentation issue.

Comment 14. Some descriptive statistics of the data would be helpful. It can stay in the supplementary information but it should contain a table with summary statistics.

References

Battiston S, Gatti DD, Gallegati M, Greenwald B, Stiglitz JE. Credit chains and bankruptcy propagation in production networks. *Journal of Economic Dynamics and Control*. 2007 Jun 1;31(6):2061-84.

Vitali, S., Glattfelder, J.B. and Battiston, S., 2011. The network of global corporate control. *PLoS one*, 6(10), p.e25995.

Point to point response on the reviewer comments:

Reviewer #1

I read with great interest the paper "Inequality in economic shock exposures across the global firm-level supply network" by Chakraborty et al.

I believe this paper is a novel and interesting example of how the recent methodological advancements in complex network theory can be fruitfully applied to the specific case of study of economics.

The paper analyses the structure of the chain of production at various levels culminating in the structure of production at the continental size. Similar to previous work done for financial systems they are able to measure the risk exposure across the different levels.

In my view, the main result of the paper is that such exposure cannot be reasonably computed without taking into account the network effects in the system. Similarly to the financial case, non considering such an effect will lead to a serious underestimation of the risk.

This paper fills a gap in the study of economic systems where network effects in the case of chains of production have been not fully considered. The methodology used is sound and reproducible, I have one question though when the authors consider the regional block structure in Fig.2.

One typical model for such things is the gravity model so that countries and/or firms have a tendency to trade with counterparts nearby so that one can consider the distance between the two places as an approximate measure of the unlikeness of establishing a business relationship. It seems from the data (I might be wrong) that if it were so there should be different "gravity models" for any continent, making therefore such hypothesis not very plausible,

Could the author comment on this specific point, that is of great importance for the community?

We thank Reviewer #1 for the time and effort put into reviewing our manuscript. Gravity models are the go-to approach to model flows between geographic entities, such as migration or—in our case— trade. In fact, we think that the gravity model is able to explain most of the structure that we observe and show in Fig. 2. To do so, we now fit a simple gravity model and present it in the SI Text 7.

For the number of links between countries, the gravity model works quite well. We find that it explains 68% of the variance in the number of links between countries. For the *Expected Exposure*, E^{cd} , between two countries the gravity model explains 44% of the variance, but has an unexpected result regarding the exponents of the in- and outflow of risk. The outgoing risk has a very small exponent and the ingoing exposure has a negative exponent. However, this result confirms our analysis from Fig. 3a; countries with large total exposure don't typically receive large exposures, but many small exposures E_{cd} . The discrepancy between the 'typical' gravity model explaining the links and the 'anomalous' model for risk

exposure highlights the main finding of our paper that risk spreads differently from regular trade flows.

To address the point raised by the reviewer about “different gravity models for any continent” we can inspect the E^{cd} reconstructed from the gravity model, shown below. The method is able to recreate the diagonal blocks, even if we fit the same gravity law for all continents.

For the rest I found the paper novel, well written and valuable as a contribution in the field and I have no hesitation in suggesting publication for it

We thank Reviewer #1 again for taking the time and the positive feedback.

Reviewer #2

Review report

Title: Inequality in economic shock exposures across the global firm-level supply network

The paper investigates an important and extremely timely questions in the current fastly evolving geopolitical context. The paper is very well written and clear. The methodology is solid and transparent. I have some points of concern, as described in general comments. I believe addressing these points is feasible. However, it requires some substantial work.

We thank Reviewer #2 for the time and effort put into reviewing our manuscript and the extensive feedback. We adapted the manuscript at the respective locations and added several SI Texts to address the points raised.

General comments:

1. given that this is not an empirical analysis of losses actually occurred but results are based on a model of possible realizations of losses, then the authors should convince the reader why the relation highlighted in the paper between expected loss and network position and/or GDP per capita can be considered correct or relevant, or why/how is it useful from a perspective of scientific understanding and validation. I do think that the exercise carried out by the authors is useful, but I also think it is necessary to explain clearly why this is the case.

We agree with reviewer 2 that it's important to convince the reader why our findings are scientifically and practically useful. In the introduction we emphasize the value of this study for the scientific understanding of global systemic risk flows (p. 3, line 162). Also in the introduction we now explain better that our model starts from first principles and actually uses no parametrization (p.3, line 182). Of course we have implicit parameters and assumptions and to evaluate them we added SI Texts 2 and 3, where we re-calibrate the microscopic mechanism presented in our model using empirical results from the literature.

2. The analysis uses empirical data of supply networks. However, the magnitude of the effects estimated depends highly (as far as I can understand) on non-validated parameters (e.g. how much important a supplier loss is for a given company). Indeed, in absence of frictions, delays, inertia or other, a company could recover 100% from the shock in very short time. The cascades that the authors find only emerge, I suspect, in presence of delays or non-linearities. This was the result from Battiston et al. 2007 and I would be surprised there can be a different result here. Hence, the effects that the paper estimates entirely depends in magnitude on the recovery capabilities, which could be measured empirically, in principle.

I suggest the authors to discuss how to carry out a validation of the model. Even if a full validation is not necessary and out of scope here, a partial validation could be attempted using data on the extreme/catastrophic events mentioned in the introduction. Another piste I suggest is to try and use existing empirical results to validate the microscopic mechanisms at work in this model. What matters is to convince the readers that the model could be validated and that it gives reasonable magnitude of estimates in a sample for which data is available.

If the success of the validation exercise is limited, a sensitivity analysis should be conducted to show how much the results are dependent on parameters related to the ability of firms to recover and also wrt to the PD of the firms. Currently, the PD is assumed the same for all firms, which is not very realistic. In reality this is going to be sector specific and time varying. What of these are details and what is important for our understanding?

The systemic risk we associate with a firm is actually not taking realistic time scales into account, but rather runs the shock spreading algorithm using an 'internal time' until all firms that could be possibly affected have been affected. This is of course far from realistic, but should give an estimate of how many firms could *in principle* be affected by the failure of a firm. Using such simplifying assumptions is common in the definition of systemic risk indices, see for example (Battiston et al. 2012, Fujiwara et al. 2016, and Diem et al. 2022). On the

one hand this is due to data limitations and on the other hand simplifying assumptions are helpful with regards to computing time.

We completely agree with the point Reviewer #2 raises here; given that we're using a relatively simple shock spreading model, we need to discuss how realistic our model is and take some effort to validate it.

First, to make it more clear to the reader what kind of effects we model and what is not modeled, we added a paragraph in the introduction to clarify what our algorithm does and does not (p. 3, line 182).

Second, we added SI Text 2 where we follow the second piste proposed by Reviewer #2, and review the literature on economic shock propagation along supply chains to validate our microscopic mechanisms.

Third, in the discussion we added three sentences where we discuss how our model could be validated in greater detail (p. 7, line 556).

Fourth, added a discussion of the effects of heterogeneous PDs in SI Text 3. We also improved our presentation of the definition of E^{cd} to differentiate between the *average exposure* and the *expected loss*, taking the probabilities of default into account or not, respectively.

Further, we introduce an 'effective replacement' mechanism and show that our results are robust to it, see SI Text 2.

3. The results for E^{cd} in Figure 2 seem smaller than 10^{-2} . These are small numbers. Why does it matter in economic terms?

How do these figure depend on assumptions? See point 2.

The values for E^{cd} are indeed small, but represent the *average exposure* per firm we expect in country c after a random firm failing in country d , so it can be actually sizable for a single firm failure. On the one hand, these numbers can get large, if there would be, for example, a systemic event in country c affecting many firms. On the other hand, large values in E^{cd} are often driven by several firms causing large cascades in d . We added a note in the discussion to clarify how the numbers in E^{cd} need to be interpreted (p. 6, line 432).

4. Related to point 2: I would suppose that the results are quite sensitive to how the parameters of the model are calibrated, e.g. the level of vulnerability of firms to the shock of disruption of one supplier. If this is the case, then what is the value of the results? Is the magnitude what matters here or something else? What is robust to those calibration details that we can learn from the paper?

In response to this point and the points above, we now added SI Texts 2 and 3, testing for the robustness of our results to different calibrations of (i) the level of vulnerability of firms to supply shocks and (ii) heterogeneous probabilities of default of the firms. Both robustness checks show that the main results of our paper, the block structure of E^{cd} and the negative correlation with GDP_{pc} , hold. Originally, we didn't put much emphasis on the magnitude of the values in E^{cd} . We are thankful to Reviewer #2 for his/her comments, because it is now clearer how the magnitude of our values can be interpreted and how robust our qualitative results are to different calibrations.

Specific comments:

Comment 1. Passage “Previous studies also argued that international production can have negative effects.”. Wording: the term “international production” seems not correct or unclear. Authors maybe mean the “outsourcing or delocalization abroad of domestic production”.

This was indeed what we meant. We changed the formulation.

Comment 2. Passage “In a similar study, the effect of the Great East Japan Earthquake has been estimated to have caused a significant 0.47 percentage point decline in Japan’s real GDP growth [23]. “ This sentence consider the same event as the previous sentence but difference between numbers is unclear. I suppose, 2.4% drop is a short-term shock, which this is a persistent reduction of growth rate. Please clarify. Please also address the fact that the expression “Great East Japan Earthquak” is repeated three times in one paragraph.

We thank Reviewer #2 for pointing this out. Partly, the difference can be explained by the fact that Inoue & Todo use an Agent Based Model, while Clarvalho et al. use a general equilibrium model. The different estimation is due to different calibrations of these models. We now mention the difference between the models in the text (p. 1).

Comment 3. Introduction: The stream of works on the ambiguous effects of production networks could be put into the broader context of the global architecture of economic networks. In this regard, the first large scale empirical study of the global economic network is Vitali et al. 2011 PLOS-ONE, although it covers a different type of networks, namely ownership, which is of course different but also related to production networks. One similarity between the two papers is that they look at firm level networks as opposed to sector-level.

We thank Reviewer #2 for pointing out that we didn’t contextualize our study within the economic-networks literature. We added a sentence in the introduction (p. 2, line 98).

Comment 4 on passage “The potential of a local disruption, e.g. the default of a single bank, causing a system-wide large disruption in a financial network (financial crisis) is called its systemic risk.”. I suppose the authors mean to say that this is its systemic impact not risk. A systemic event is usually an event with large losses at the system level. The authors should make consistent definitions of the expressions involving “risk”. If risk is just a probability, then use probability instead. If “risk” is an expected loss please explain. In financial markets “risk” means volatility. So “risk” means different things to different audiences and it must be defined.

Thank you for this comment. Our definition & formulation is in line with previous work, e.g. (Diem et al. 2022), where the impact a firm *could* have on the whole system is called its ‘systemic risk’. However, we adapted our formulation in the paragraph and throughout the paper to be more clear that we mean the “systemic risk a firm poses to a system.”

Comment 5 on passage: “Illustrative examples of shock spreading on the single firm level are the global shortage in hard drives subsequent to the 2011 flood in Thailand

[39], or the ongoing shortage in computer chips [40, 41].” Please replace ongoing with a reference to 2022, for future readability.

Thank you for the remark, we added a reference to 2021/2022 (p.2, line 93)

Comment 6 on passage: “In the case of a supply shock, we call this mechanism the downstream propagation of shocks or the downstream cascade. The same logic applies to demand reductions that propagate upstream, or equivalently, cause an upstream cascade (not shown).” This is indeed a very relevant concept, which was previously studied in Battiston et al 2007 in the context of a model of production network (coupled with credit). The authors showed conditions (analytically under homogeneity assumptions) for the occurrence of either one type of cascades or both.

We are thankful for the pointer to this paper. It is indeed very relevant and we added a reference to it.

Comment 7 on passage: “For example, if we rewire the outgoing link of firm 5, such that it now supplies firm 2 instead of firm 1, we preserve the number of links between countries Acd, but the exposure of countries A and C to B would be substantially larger than compared to the situation shown in Fig. 1.” Not very clear. The figure refers to the case with no rewiring. How do we see that after rewiring the difference would be big. The concept is important and probably worth showing explicitly.

We are grateful for this suggestion and added a figure comparing the cascade of firm 5 in the original and the rewired network to make the difference more easily accessible. Due to the limited space available we placed the figure in SI Text 4.

Comment 8 on Fig 2. The term “Expected exposure from country ... to country”: the concept is clear but there is an English problem I suppose the common expression is exposure of a firm/country to another firm/country. I guess you can say impact from A to B but you cannot say exposure from A to B.

Of course links or shocks can be from one firm/country to another, but not exposure. We corrected the error.

[comment 9 is missing]

Comment 10 on Passage: “80% of down population that are least exposed to risk are exposed to only around 10% of all risk, or vice-versa, 20% of the most exposed population carries around 90% of the risk.” The passage could be revised to make easier reading the figure, e.g. referring to color of curve.

Thank you, we now mention the colors of the respective curves in the main text.

Discussion

Comment 11 on passage: “A regression analysis solves the puzzle and we find that GDP per capita correlates negatively ($r = 0.52, p < 10^{-8}$) and, hence, poor countries

have a higher total exposure. “ . I am confused now, is the passage referring to the same regression presented earlier in Results, or is it a new exercise?

This is indeed the same exercise as presented in the results. We removed this repetition in the process of streamlining the Discussion in response to Comment 13.

Comment 12 on passage: “With the currently available data we find no evidence for a “risk premium” in the sense that higher exposures would co-occur with significantly higher economic growth rates.” The concept of risk premium appearing here is not explained. The term is used in finance and economics to indicate a differential in yield or return that market participants demand from holding a security in compensation for its risk. The analogy here is not clear and needs to be explained. I am not convinced it is a necessary or useful metaphor, since there is no holder of a contract here that could demand compensation for risk.

We agree with the Reviewer. We rephrased the text without mentioning “risk premium”.

Comment 13. The last two pages of discussion are a bit confusing to me. They overlap with the section results and repeat some of the results. I suggest to streamline and remove redundancy. The section could be shortened substantially, unless I missed that there is important new information and then maybe there is a presentation issue.

Thank you for this feedback. To improve the readability and clarity of our manuscript we shortened and streamlined the recapitulation of our results in the Discussion.

Comment 14. Some descriptive statistics of the data would be helpful. It can stay in the supplementary information but it should contain a table with summary statistics.

We agree that some summary statistics are helpful to contextualize the results. We added a table with descriptive statistics of the network in SI Text 13.

References

Battiston S, Gatti DD, Gallegati M, Greenwald B, Stiglitz JE. Credit chains and bankruptcy propagation in production networks. Journal of Economic Dynamics and Control. 2007 Jun 1;31(6):2061-84.

Vitali, S., Glattfelder, J.B. and Battiston, S., 2011. The network of global corporate control. PloS one, 6(10), p.e25995.

We thank Reviewer #2 again for the valuable feedback.

References:

Diem, C., Borsos, A., Reisch, T., Kertész, J., & Thurner, S. (2022). Quantifying firm-level economic systemic risk from nation-wide supply networks. Scientific reports, 12(1), 7719.

REVIEWER COMMENTS

Reviewer #1 (Remarks to the Author):

My concerns have been answered in a satisfying way. I do not have any other objection

Reviewer #3 (Remarks to the Author):

The paper is interesting and well written. The topic is very important and timely. The claims made in the paper are relevant. However, I have some major and minor remarks that I list below. They mainly concern methodological issues and the extent to which the quality of data employed in the study robustly allows one to derive the main results discussed in the paper.

Major remarks

1) The paper makes several bold claims, namely: a) exposures to other countries are highly structured on a regional level; b) high income countries expose a large fraction of the globe to economic losses; c) low income countries are disproportionately strongly affected by high exposure values. The first result is validated from observing the existence of a very simple block-diagonal structure in the matrix of E_{cd} , while the second and third claim is derived from the same matrix above, now grouped by income level. Furthermore the paper suggests that: d) higher exposure is not positively correlated with higher gains in GDP growth rates in recent decades, which stems from a regression analysis; and e) global economic exposure is distributed more unequally than income per capita (using Gini coefficients).

My main concern is about whether the quality of available data, on the one hand, and some methodological assumptions made in the paper, on the other, allows one to robustly sustain the main claims made in the paper.

1.1 On the data-quality side, there seems to be a big bias in terms of representativeness of the data. The dataset features about 230K firms, from a likely number that is of the order of several millions. This poses a representativeness issue, as only a tiny percentage of the universe is represented in the sample employed. The same argument applies to the number of observed connections in the network, which tends to average at about 2-3 incoming and outgoing connections per firm. This seems to be a very peculiar snapshot of the original, true, network of connections. The claims made in the paper should in my view take into account this representativeness issue.

1.2 From the methodological perspective, using a DebtRank based model without weights associated to the links (binary network) — and the lack of an analysis of their importance — means that we do not really know how much those links actually diffuse production losses (i.e., firms not being able to produce due to a lack in inputs). More generally, underestimating the importance of certain links and overestimating the importance of others may substantially bias the results in terms of the global structure of the exposure between countries, especially if link-weight distributions are highly skewed (as it often happens in real-world economic networks due to size effects). Furthermore, using degrees as proxies for firm size is not per se a bad idea. However, due to the data-quality issue mentioned in point 1.1, it is not clear to me whether the observed degree sequence is in any way representative of the true structure of the exposure network.

Minor comments:

1. Regression analyses (see SI, 7 and 8) should be improved. Using exports/imports per capita to predict GDP per capita introduces strong endogeneity and omitted-variable bias issues.
2. lines 80, 516, 537, 589 - something wrong with reference latex

3. lines 322-324 - language: countries exporting distress seems a bit of an excessive wording, maybe they are central in propagating distress or something similar
4. line 556 - typo: "we do are not"
5. Some results hint at some size effects being at play. For example, the fact that EU countries are those exposing the most (to most countries), or the fact that Fig. 2a and 2b. seem quite similar; or also the fact that one can observe "stripes" in Fig. 2a (and in Fig. S16 - where richer US states seem to expose the most, to the majority of states). A discussion on this point should be made in the paper.
6. I believe that studying a "premium" on the production risk is an interesting idea; however, I don't find GDP growth a convincing variable per se. I would suggest complementing the analysis with some other measure(s) that might be closer in spirit to the upside and downside of "offshoring" or production more generally, e.g., volatility of exports, size/rate of Foreign Direct Investment (FDI), etc.

Response to reviewers

Reviewer #1

My concerns have been answered in a satisfying way. I do not have any other objection

We thank Reviewer #1 for taking the time and the positive feedback.

Reviewer #3

The paper is interesting and well written. The topic is very important and timely. The claims made in the paper are relevant. However, I have some major and minor remarks that I list below. They mainly concern methodological issues and the extent to which the quality of data employed in the study robustly allows one to derive the main results discussed in the paper.

Major remarks

1) The paper makes several bold claims, namely: a) exposures to other countries are highly structured on a regional level; b) high income countries expose a large fraction of the globe to economic losses; c) low income countries are disproportionately strongly affected by high exposure values. The first result is validated from observing the existence of a very simple block-diagonal structure in the matrix of E_{cd} , while the second and third claim is derived from the same matrix above, now grouped by income level. Furthermore the paper suggests that: d) higher exposure is not positively correlated with higher gains in GDP growth rates in recent decades, which stems from a regression analysis; and e) global economic exposure is distributed more unequally than income per capita (using Gini coefficients).

My main concern is about whether the quality of available data, on the one hand, and some methodological assumptions made in the paper, on the other, allows one to robustly sustain the main claims made in the paper.

1.1 On the data-quality side, there seems to be a big bias in terms of representativeness of the data. The dataset features about 230K firms, from a likely number that is of the order of several millions. This poses a representativeness issue, as only a tiny percentage of the universe is represented in the sample employed. The same argument applies to the number of observed connections in the network, which tends to average at about 2-3 incoming and outgoing connections per firm. This seems to be a very peculiar snapshot of the original, true, network of connections. The claims made in the paper should in my view take into account this representativeness issue.

We thank Reviewer #3 for the comment. It is indeed important to address the issue of data quality and representativeness. We added a new SI section (SI Text 12) where we describe a series of tests and checks that investigate the data representativeness issue raised above.

First, we investigate the robustness to missing nodes by (a) randomly sampling 10% of the firms in the global supply network with a uniform probability, (b) sampling the largest 10% of the firms, and (c) sampling the largest 1% of the firms, and computing the respective induced subgraphs. Then we compare the results of the shock spreading algorithm on the induced subgraphs to the results on the full network, to learn about potential uncertainties or biases. We find that the results of the dynamics on the small network are highly correlated to the results on the full network, but the size of the damage is potentially overestimated.

Second, we perform a similar exercise to learn about the robustness with respect to missing links. It is safe to assume that the links contained in the S&P Capital IQ database are large, important links. We argue so, because the database collects public information on buyer-supplier relations, which is typically only available for large, recurring supplier relationships. In fact, a large part of the data is based on '10-K filings' in the US, in which (among other information) large firms have to declare all customers that represent more than 10% of its sales. To test the robustness of our results on a weighted network, we brought a new co-author on board, who provided us with the supply network of Ecuador. The supply network of Ecuador is reconstructed from high-quality value added tax (VAT) data and considerably denser than the global supply network. We generate two networks, one by (i) removing all links below \$100,000 annual transaction volume, thereby reducing the average degree to a density comparable to the one we observe for the global supply chain network; and a second network (ii) where we remove all links that represent less than 10% sales share of the supplier. For the first network (i) we find that DebtRank on the full and filtered network are highly correlated, and that its values on the filtered network are larger than on the full network. On the second network (ii) DebtRank before and after filtering still correlates well with a Pearson $r = 0.58$.

1.2 From the methodological perspective, using a DebtRank based model without weights associated to the links (binary network) — and the lack of an analysis of their importance — means that we do not really know how much those links actually diffuse production losses (i.e., firms not being able to produce due to a lack in inputs). More generally, underestimating the importance of certain links and overestimating the importance of others may substantially bias the results in terms of the global structure of the exposure between countries, especially if link-weight distributions are highly skewed (as it often happens in real-world economic networks due to size effects). Furthermore, using degrees as proxies for firm size is not per se a bad idea. However, due to the data-quality issue mentioned in point 1.1, it is not clear to me whether the observed degree sequence is in any way representative of the true structure of the exposure network.

It is certainly true that we should address the bias introduced by using the binary instead of the weighted supply network. Initially, with the original S&P data this was not possible, but now with the Ecuadorian supply network available we are in the exciting position that we can perform a similar analysis as described above. In SI Text 12 we now show that results on the binary and weighted network are highly correlated. We further show that for the network where we remove all links below \$100,000 annual transaction volume the correlation between the weighted and unweighted DebtRank actually increases.

The Ecuadorian VAT data allows us to also address the second point raised by Reviewer #3, the representativeness of a node's degree for its sales, especially in the presence of missing links. In SI Text 12 we conclude by showing that degree and sales are highly correlated on the unfiltered network, and that this correlation increases substantially if the network is thresholded.

Minor comments:

1. Regression analyses (see SI, 7 and 8) should be improved. Using exports/imports per capita to predict GDP per capita introduces strong endogeneity and omitted-variable bias issues.

We revised SI Text 8 and removed exports/import per capita from the regression. This indeed introduced a strong omitted-variable bias and should be avoided. Thank you for bringing this to our attention.

We don't think that the same issue exists for SI Text 7, there we follow the standard literature and explain (a) the number of links with each country's in- and out-degree and their distance and (b) the exposure with each country's in- and out-flow of exposure.

2. lines 80, 516, 537, 589 - something wrong with reference latex

Thank you for pointing this out. The error occurred when updating a reference from the preprint to the published paper version.

3. lines 322-324 - language: countries exporting distress seems a bit of an excessive wording, maybe they are central in propagating distress or something similar

We wanted to emphasize the direction of how distress 'flows', but reformulated the sentence to the more precise "cause distress"/"exposed to distress".

4. line 556 - typo: "we do are not"

Thank you for pointing this out.

5. Some results hint at some size effects being at play. For example, the fact that EU countries are those exposing the most (to most countries), or the fact that Fig. 2a and 2b. seem quite similar; or also the fact that one can observe "stripes" in Fig. 2a (and in Fig. S16 - where richer US states seem to expose the most, to the majority of states). A discussion on this point should be made in the paper.

This is indeed a very interesting point for the discussion. We're trying to control for this by the normalization we use, but a certain size effect could remain because larger countries can sustain larger firms. We added a few sentences (line 432ff) in the discussion hinting towards this and an additional SI Text where we normalize E^{ω} with a country's total degree instead of its number of firms, showing the robustness of our results to a different size proxy.

6. I believe that studying a "premium" on the production risk is an interesting idea; however, I don't find GDP growth a convincing variable per se. I would suggest complementing the analysis with some other measure(s) that might be closer in spirit to the upside and downside of "offshoring" or production more generally, e.g., volatility of exports, size/rate of Foreign Direct Investment (FDI), etc.

Thank you for this comment. We agree that the signal of a 'risk premium' in GDPpc growth might be obscured by other factors, and we should investigate quantities related more directly to international trade and/or offshoring. In SI Text 9 we study the correlation of downstream exposure with the inflow of FDI per capita and don't find any significant correlation.

We also investigated the volatility of exports. For this quantity we, however, expect to see a positive correlation with the exposure to shocks because it is a proxy for exactly these. Indeed, as shown in the plot below, downstream exposure and the standard deviation of log-returns over 20 years are significantly positively correlated, Pearson's $r = 0.21$, $p=0.03$. We chose not to include this result in SI Text 9, because it doesn't really relate to a potential 'premium' systemic risk.

REVIEWERS' COMMENTS

Reviewer #3 (Remarks to the Author):

My remarks have been addressed in a satisfying way. I do not have any further comment.